# Dynamic Masking and Auxiliary Hash Learning for Enhanced Cross-Modal Retrieval

**Shuang Zhang**[1,2], **Yue Wu**[1], **Lei Shi**[3,*], **Yingxue Zhang**[4], **Feifei Kou**[5,6],
**Huilong Jin**[1], **Pengfei Zhang**[7], **Meiyu Liang**[5], **Mingying Xu**[8]

[1]College of Engineering, Hebei Normal University
[2]Hebei Provincial Key Laboratory of Information Fusion and Intelligent Control
[3]State Key Laboratory of Media Convergence and Communication, Communication University of China
[4]College of Computer and Cyber Security, Hebei Normal University
[5]School of Computer Science (National Pilot School of Software Engineering), BUPT
[6]key Laboratory of Trustworthy Distributed Computing and Service, BUPT, Ministry of Education
[7]School of Computer Science and Engineering, Anhui University of Science of Technology
[8]School of Artificial Intelligence and Computer Science, North China University of Technology
*Corresponding author: `leiky_shi@cuc.edu.cn`

## Abstract

The demand for multimodal data processing drives the development of information technology. Cross-modal hash retrieval has attracted much attention because it can overcome modal differences and achieve efficient retrieval, and has shown great application potential in many practical scenarios. Existing cross-modal hashing methods have difficulties in fully capturing the semantic information of different modal data, which leads to a significant semantic gap between modalities. Moreover, these methods often ignore the importance differences of channels, and due to the limitation of a single goal, the matching effect between hash codes is also affected to a certain extent, thus facing many challenges. To address these issues, we propose a Dynamic Masking and Auxiliary Hash Learning (AHLR) method for enhanced cross-modal retrieval. By jointly leveraging the dynamic masking and auxiliary hash learning mechanisms, our approach effectively resolves the problems of channel information imbalance and insufficient key information capture, thereby significantly improving the retrieval accuracy. Specifically, we introduce a dynamic masking mechanism that automatically screens and weights the key information in images and texts during the training process, enhancing the accuracy of feature matching. We further construct an auxiliary hash layer to adaptively balance the weights of features across each channel, compensating for the deficiencies of traditional methods in key information capture and channel processing. In addition, we design a contrastive loss function to optimize the generation of hash codes and enhance their discriminative power, further improving the performance of cross-modal retrieval. Comprehensive experimental results on NUS-WIDE, MIRFlickr-25K and MS-COCO benchmark datasets show that the proposed AHLR algorithm outperforms several existing algorithms.

## 1 Introduction

In the field of information retrieval, although traditional unimodal data representation is efficient in processing single-type data, it is difficult to capture data associations and semantic details in the face of increasingly popular cross-modal data. Cross-modal retrieval[1][2][3] as an emerging solution, it effectively bridges the gap between heterogeneous modalities by establishing connections

39th Conference on Neural Information Processing Systems (NeurIPS 2025).

between different data types. In recent years, cross-modal hashing retrieval[4][5][6][7] has attracted widespread attention due to its advantages of fast retrieval and efficient storage. It uses hashing technology to convert high-dimensional data into low-dimensional binary hash codes, thereby reducing computational complexity and storage requirements while retaining semantic information.

Currently, some scholars have proposed a variety of new cross-modal hashing retrieval methods. Neural network technologies, such as convolutional neural networks (CNNs) and generative adversarial networks (GANs), have been widely used in cross-modal hashing retrieval. CNN can effectively extract semantic information from images through its powerful feature extraction capabilities, while GAN generates robust hash codes through adversarial training of generators and discriminators, thereby improving the performance of cross-modal retrieval. In addition, large language models (LLMs) have also been introduced into cross-modal hashing retrieval, which enhance the semantic representation of text modalities through their powerful natural language processing capabilities, thereby improving the accuracy of cross-modal matching.

Although many methods have achieved good results in the field of cross-modal hashing retrieval, they still face some challenges. Due to the huge semantic gap between different modalities [8][9] often leads to inconsistent cross-modal representations, many noncritical information or noise[10] may affect the matching accuracy, resulting in similar images and texts being mismatched[11]. Secondly, when processing features, traditional hash layers often ignore the importance differences between different channels [12], which can lead to insufficient capture of key information and difficulty in effectively suppressing noise and redundant information. In addition, when hash codes are generated, they usually rely on a single optimization goal, which may lead to insufficient performance of hash codes in cross-modal matching.

To effectively address these challenges, we proposed a method called auxiliary hashing learning (AHLR). It significantly improves feature extraction and alignment capability by introducing a dynamic mask mechanism. Specifically, the dynamic mask can automatically identify and weight key information in the image and text during the training process, effectively improving the accuracy of matching of cross-modal features. In addition, we also constructed an auxiliary hashing layer that can adaptively weight the features of each channel, thereby solving the problem of channel information imbalance, while enhancing the ability to capture key information and effectively suppressing noise interference. Finally, by introducing a contrastive loss function, minimizing the distance between similar samples, and maximizing the distance between heterogeneous samples, the distinguishing ability of hash codes in cross-modal retrieval is effectively enhanced, thereby improving the retrieval accuracy. The main contributions of this paper are as follows:

- We propose a dynamic masking and auxiliary hash learning (AHLR) method for cross-modal retrieval, which can effectively enhance feature extraction and alignment capabilities, generate more detailed hash codes, and improve the accuracy of cross-modal hashing retrieval.

- We introduce a dynamic masking mechanism to automatically select key information from images and text and weight it, thereby improving the accuracy of feature alignment and matching.

- We design an adaptive auxiliary hash learning cross-modal module that can adaptively weight the features of each channel, enhancing the retention of key information. Moreover, we introduce the contrast loss function to distinguish the similarity and heterogeneity of the samples and improve cross-modal semantic consistency.

- Extensive experiments on three benchmark datasets show that our AHLR outperforms state-of-the-art baselines, demonstrating clear performance advantages.

## 2 The Proposed Method

### 2.1 Notation and problem definition

For ease of understanding, we first introduce the following notation used in this article. Assume that there is a training dataset $\mathbf{O} = \{(\mathbf{X}_i, \mathbf{X}_t) \,|\, i \in [1, N], t \in [1, N]\}$, where $\mathbf{X}_i$ represents the i-th image sample, $\mathbf{X}_t$ represents the t-th text sample and $N$ represents the number of samples. The size of the image is defined as $\mathbf{X} \in \mathbb{R}^{H \times W \times C}$, where H, W, and C represent the height, width,

and number of channels of the image, respectively. $\mathbf{F}_I$ and $\mathbf{F}_T$ represent the features after the extraction of the image and text. The dynamic attention mask is used to process the features of the image and text data, and the generated dynamic mask matrix is defined as $\mathbf{M}$. When $\mathbf{M} = 0$, the model calculates the attention weight normally, and when $\mathbf{M} = -\infty$, the model focuses only on key information. The auxiliary hash layer performs channel dimensionality reduction and weighting processing on the features to obtain the final embedded features $f_I^*$ and $f_T^*$. The binary hash code $\mathbf{B}_i$ is generated by the maximum probability selection method. The final hash vector is represented as $\mathbf{H} = (\mathbf{B}_1, \mathbf{B}_2, \ldots, \mathbf{B}_K) \in \{0, 1\}^{\mathbf{K}}$ and K represents the length of the hash code.

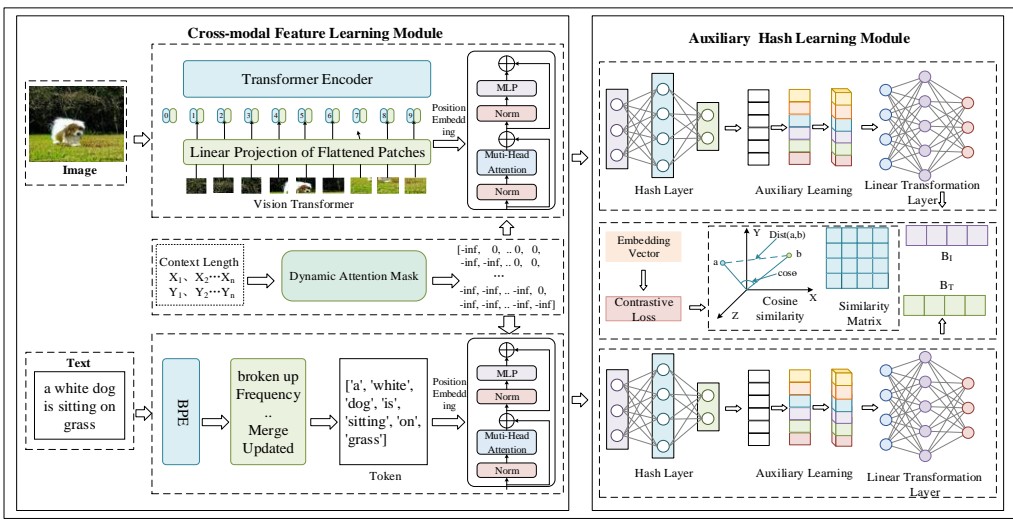

Figure 1: The AHLR framework consists of two main modules: 1) a cross-modal feature learning module that extracts features from images and texts. By introducing a dynamic attention mask, it automatically selects and weights the key information in images and texts according to the length of the input sequence; 2) an auxiliary hash learning module that adaptively weights the features of each channel, optimizes the feature representation to assist in the generation of hash codes, and introduces a contrast loss function to enhance the distinguishing ability of hash codes.

## 2.2 Framework Architecture of AHLR

The AHLR framework in Figure1 contains two modules: Cross-modal Feature Learning Module and Auxiliary Hash Learning Module. For cross-modal feature learning, the Vision Transformer is employed as the image feature extractor, after which the features are fused via a Transformer to obtain the final image representation. For the text modality, Byte Pair Encoding (BPE) is first applied to segment the input text, followed by a Transformer-based fusion process to generate the corresponding text representation. In order to adapt to input sequences of different lengths, the model introduces a dynamic attention mask mechanism to automatically generate the corresponding mask matrix, so that the model can effectively ignore irrelevant information and focus on key features. Traditional attention mechanisms assign continuous soft weights to all tags or image patches. However, even information with minimal contribution to the task is retained rather than discarded. In contrast, the proposed dynamic masking mechanism adaptively selects semantic key information from both image and text modalities during training and assigns appropriate weights. This process effectively suppresses modality redundancy and irrelevant regions, thereby enhancing semantic consistency across modalities, strengthening feature alignment, and ultimately improving retrieval efficiency. In the hash code learning part, an auxiliary hash layer is designed to enhance the representation ability of the hash code by adaptively weighting the features of each channel, and a contrast loss function is used to strengthen the feature aggregation between similar samples, while pulling away the hash representation of dissimilar samples to improve the retrieval performance of the model.

## 2.3 Cross-modal Feature Learning Module

In the AHLR model, the cross-modal feature learning module is mainly divided into two steps, namely image feature learning and text feature learning.

### 2.3.1 Image Feature Learning

First, use the Vision Transformer (ViT)[13] to extract deep features from the input image. Given an input image of size $\mathbf{X} \in \mathbb{R}^{H \times W \times C}$, it is split into patches of size $P \times P$, and the total number of patches generated $\mathbf{N}$ is represented as follows:

$$\mathbf{N} = \frac{H}{P} \cdot \frac{W}{P} = \frac{H \cdot W}{P^2} \tag{1}$$

where, H, W, and C represent the height, width, and number of channels of the image, respectively. Each patch is flattened into a vector to form a matrix representation $\mathbf{X}_{patch}$. The specific process is as follows:

$$\mathbf{X}_{patch} \in \mathbb{R}^{N \times (P^2 \cdot C)} \tag{2}$$

The shape of the matrix representation is $P^2.C$, which represents the feature dimension of each patch after flattening. Subsequently, it is mapped to a feature space of dimension D through the linear transformation. The mapping process is as follows:

$$\mathbf{F}_i = \mathbf{X}_{patch}\mathbf{W}_E + \mathbf{b}_E, \mathbf{W}_E \in \mathbb{R}^{(P^2 \cdot C) \times D}, \mathbf{b}_E \in \mathbb{R}^D \tag{3}$$

where, $\mathbf{W}_E$ is the linear transformation matrix and $\mathbf{b}_E$ is the bias term. Then, to maintain spatial information, a position code P is added to each patch. The specific input of the Vision Transformer is as follows:

$$\mathbf{F}_i^{'} = \mathbf{F}_i + P, P \in \mathbb{R}^{N \times D} \tag{4}$$

The encoded patch features are input into the Transformer. The dynamic attention mask sets the corresponding mask matrix according to the sequence length of the image input. The definition of the dynamic attention mask matrix $\mathbf{M}$ is as follows:

$$\mathbf{M}_{ij} = \begin{cases} 0 & \text{if } j \leq i \\ -\infty & \text{if } j > i \end{cases} \tag{5}$$

When $\mathbf{M}_{ij} = 0$, this means that the model normally calculates the attention weight. When $\mathbf{M}_{ij} = -\infty$, the attention score of this position will be completely blocked, which will make the model focus on the key parts while avoiding the interference of invalid information. The dynamic attention mask matrix restricts each token to attend only to its current and previous positions. Although the mask matrix is determined by the length of the input sequence and does not directly contain semantic information, during the training process, the model can model global and local information through the multi-head attention mechanism and feed-forward network, thereby gradually learning key information under the mask restriction. Subsequently, the global and local information of the image is captured through the multi-head self-attention mechanism and feedforward network. The specific formula is as follows:

$$\mathbf{Q} = \mathbf{F}_i^{'}\mathbf{W}_Q, \mathbf{K} = \mathbf{F}_i^{'}\mathbf{W}_K, \mathbf{V} = \mathbf{F}_i^{'}\mathbf{W}_V \tag{6}$$

$$\text{Attention}(\mathbf{Q}, \mathbf{K}, \mathbf{V}) = softmax\left(\frac{\mathbf{Q}\mathbf{K}^T}{\sqrt{d_k}} + \mathbf{M}\right)\mathbf{V} \tag{7}$$

$$\mathbf{X}_i = Concat(\text{Attention}_1, \dots \text{Attention}_h)\mathbf{W^O} \tag{8}$$

where, $\mathbf{W}_Q$, $\mathbf{W}_K$ and $\mathbf{W}_V$ represent the transformation matrices of query, key and value, respectively, $d_k$ is the dimension of the key vector, and $\mathbf{W^O}$ is the output projection matrix of multi-head attention. Finally, after the MLP layer of the Transformer encoder, the final feature representation $\mathbf{F}_I$ of the image is obtained. The specific process is as follows:

$$\mathbf{F}_I = MLP(\mathbf{X}_i) \tag{9}$$

### 2.3.2 Text Feature Learning

For text data, the model uses Byte Pair Encoding (BPE)[14] for subword-level encoding, splitting the input text into subword units, and gradually merging high-frequency subword pairs according to the frequency of occurrence of subword pairs in the text, thereby generating tokens with more text semantic information. After BPE encoding, the text representation is as follows:

$$\mathbf{F}_t \in \mathbb{R}^{L \times D} \tag{10}$$

where, L represents the length of the token sequence and D is the feature dimension. Next, add the corresponding position encoding P to the final tokens to ensure that the model can understand the order relationship in the word sequence. The process of adding position encoding is shown as follows:

$$\mathbf{F}_t^{'} = \mathbf{F}_t + P, P \in \mathbb{R}^{L \times D} \tag{11}$$

$\mathbf{F}_t^{'}$ is the final input feature after adding the position encoding. The tokens are sent to the Transformer model, the following process is similar to the image processing process, and the final feature representation $\mathbf{F}_T$ of the text is obtained. The specific process is as follows:

$$\mathbf{F}_T = MLP(\mathbf{X}_t) \tag{12}$$

## 2.4 Auxiliary Hash Learning Module

When processing features, traditional hash layers often ignore the importance differences between different channels, which may lead to insufficient capture of key information and difficulty in effectively suppressing noise and redundant information. In addition, traditional hashing methods usually rely on a single optimization goal, which may lead to poor performance of hash codes in cross-modal matching. To solve this problem, we built an auxiliary hash code generation module that is designed to generate hash codes that contain more information.

First, the linear hash layer performs a linear transformation on the input features $\mathbf{F}_I$ and $\mathbf{F}_T$, mapping the high-dimensional input of images and texts into a low-dimensional embedding space to generate a preliminary feature representation. The specific mapping process is as follows:

$$f_i = \mathbf{F}_I \mathbf{W}_I + \mathbf{b}_I, f_t = \mathbf{F}_T \mathbf{W}_T + \mathbf{b}_T \tag{13}$$

where $\mathbf{W}_I, \mathbf{W}_T \in \mathbb{R}^{D \times d}$ is the weight dimension reduction matrix, which maps the original characteristic dimension D to the embedding space of dimension d, and $\mathbf{b}_I, \mathbf{b}_T \in \mathbb{R}^d$ is the bias term. The mapped features are processed nonlinearly through the ReLU activation function. The main processing process is as follows:

$$f_I^{'} = \mathrm{ReLU}(f_i), f_T^{'} = \mathrm{ReLU}(f_t) \tag{14}$$

The auxiliary hash layer is used to adjust the channel weights of the embedded features $f_I^{'}$ and $f_T^{'}$ after dimensionality reduction, and the weights are adaptively assigned according to the global importance of each channel. The main process is to average pool the feature maps in all feature channels to obtain the global statistical information $\mathbf{S}$ of the channel. The specific process is shown as follows:

$$\mathbf{S} = \frac{1}{N} \sum_{i=1}^{N} f_j^{'}[i, :], \mathbf{S} \in \mathbb{R}^C \tag{15}$$

where $j$ can be expressed as $I$ or $T$, then $f_j^{'}$ represents the feature $f_I^{'}$ or $f_T^{'}$, the reduced dimension feature of the image and text. $N$ represents the spatial dimension of the feature. The importance weight of each channel is calculated using the fully connected layers ($FC_1$ and $FC_2$), and the global statistical information $\mathbf{S}$ is reduced and restored so that the model can capture the nonlinear relationship between channels. $FC_1$ reduces the number of channels from C to a smaller dimension (C/R), and then activates through ReLU. The process of dimensionality reduction and activation is shown as follows:

$$\mathbf{Q} = \mathrm{ReLU}(\mathbf{S}\mathbf{W}_{FC_1} + \mathbf{b}_{FC_1}), \mathbf{Q} \in \mathbb{R}^{\frac{C}{R}} \tag{16}$$

where $R$ represents the channel compression rate, which is used to control the number of channels after dimensionality reduction. Then $FC_2$ is used to restore the channels to the original number of channels $C$. The specific recovery process $\mathbf{V}$ is shown as follows:

$$\mathbf{V} = \mathbf{Q}\mathbf{W}_{FC_2} + \mathbf{b}_{FC_2}, \mathbf{V} \in \mathbb{R}^C \tag{17}$$

Finally, the weight coefficient $\mathbf{W}_i$ of each channel is obtained through the Sigmoid activation function, and the activation generates the channel weight representation as follows:

$$\mathbf{W}_i = \text{Sigmoid}(\mathbf{V}), \mathbf{W}_i \in [0, 1]^C \tag{18}$$

Each channel $C$ of features $f_I'$ and $f_T'$ is multiplied by the corresponding weight $\mathbf{W}_i$ to strengthen the representation of important channels and suppress the influence of unimportant channels. The specific process is shown as follows:

$$f_I^* = \mathbf{W}_i \odot f_I', f_T^* = \mathbf{W}_i \odot f_T' \tag{19}$$

where $\odot$ represents the channel-by-channel dot product, $f_I^*$ and $f_T^*$ represent the embedded features after dimension reduction and weighting. The embedded features $f_I^*$ and $f_T^*$ are mapped using the linear transformation layer $\mathbf{F(.)}$ to map the hash code of each bit to the binary probability distribution. The specific mapping process is shown as follows:

$$\mathbf{F}(f_x^*) = \mathbf{W}_H f_x^* + \mathbf{b}_H \tag{20}$$

where $\mathbf{W}_H$ represents the trainable weight matrix, which is mainly used to transform $f_x^*$ into a new space, and $\mathbf{b}_H$ represents the bias term. After the mapping transformation, the final hash code is determined by the calculated probability distribution $\mathbf{F}_x$. The main process is shown as follows:

$$\mathbf{F}_x = \text{softmax}(\mathbf{F}(f_x^*)) \tag{21}$$

where $x$ can be represented as an image or text. The final binary is determined by the maximum probability selection method. The hash code generation process is shown as follows:

$$\mathbf{B}_i = \begin{cases} 0 & \text{if } \mathbf{F}_i[0] > \mathbf{F}_i[1] \\ 1 & \text{if } \mathbf{F}_i[0] \leq \mathbf{F}_i[1] \end{cases} \tag{22}$$

where $\mathbf{B}_i$ is the binary hash code of the i-th bit, the length of the hash code is represented by $K$, and the vector representation of the entire hash code is shown as follows:

$$\mathbf{H} = (\mathbf{B}_1, \mathbf{B}_2, \ldots, \mathbf{B}_K) \in \{0, 1\}^K \tag{23}$$

The auxiliary hash code generation module solves the channel neglect problem of the traditional hash layer through channel weighting, enhances the expressiveness of the hash code, and makes the matching between different modal data in the model more robust.

## 2.5 Loss Function

The model introduces contrast loss into the hash code module, aiming to maximize the similarity between positive sample pairs (image and text) while minimizing the similarity between negative samples. First, the mapped image feature $f_i$ and text feature $f_t$ are represented as $\mathbf{X}_1$ and $\mathbf{X}_2$, and normalized so that the vector module length between different modalities is 1, thereby avoiding the similarity calculation deviation caused by different vector lengths. The specific normalization process is shown as follows:

$$\widehat{\mathbf{X}_1} = \frac{\mathbf{X}_1}{\|\mathbf{X}_1\|_2}, \widehat{\mathbf{X}_2} = \frac{\mathbf{X}_2}{\|\mathbf{X}_2\|_2} \tag{24}$$

$\widehat{\mathbf{X}_1}$ and $\widehat{\mathbf{X}_2}$ represent the normalized image and text embeddings, respectively. Then dot product calculation is performed to calculate the cosine similarity matrix between the image and the text. The main definition is as follows:

$$\mathbf{S}_{ij} = \frac{\mathbf{X}_1^i \cdot \mathbf{X}_2^j}{\|\mathbf{X}_1^i\|_2 \cdot \|\mathbf{X}_2^j\|_2} \tag{25}$$

Table 1: The mAP comparison results on three datasets

| Task | Method | MIRFlickr-25K | | | NUS-WIDE | | | MS-COCO | | |
|------|--------|--------|--------|--------|--------|--------|--------|--------|--------|--------|
| | | 16bits | 32bits | 64bits | 16bits | 32bits | 64bits | 16bits | 32bits | 64bits |
| I→T | DJSRH | 0.6652 | 0.6873 | 0.6987 | 0.5271 | 0.5582 | 0.6015 | 0.5257 | 0.5454 | 0.5646 |
| | JDSH | 0.7276 | 0.7426 | 0.7468 | 0.6536 | 0.6601 | 0.6900 | 0.5928 | 0.6348 | 0.6517 |
| | CDTH | 0.7317 | 0.7461 | 0.7477 | 0.6596 | 0.6613 | 0.6700 | 0.5853 | 0.6411 | 0.6573 |
| | UCCH | 0.7606 | 0.7620 | 0.7674 | 0.6718 | 0.6738 | 0.6891 | 0.6039 | 0.6249 | 0.6398 |
| | MLCAH | 0.7960 | 0.8080 | 0.8150 | 0.6440 | 0.6410 | 0.6430 | 0.5700 | 0.5620 | 0.5620 |
| | DCHMT | 0.8177 | 0.8221 | 0.8261 | 0.6711 | 0.6812 | 0.6932 | 0.6450 | 0.6331 | 0.6647 |
| | **AHLR** | **0.8203** | **0.8233** | **0.8266** | **0.6777** | **0.6884** | **0.6994** | **0.6454** | **0.6582** | **0.6797** |
| T→I | DJSRH | 0.6710 | 0.6958 | 0.7043 | 0.5575 | 0.5680 | 0.5952 | 0.5590 | 0.5591 | 0.5519 |
| | JDSH | 0.7304 | 0.7326 | 0.7481 | 0.6439 | 0.6640 | 0.6921 | 0.5888 | 0.6510 | 0.6635 |
| | CDTH | 0.7315 | 0.7464 | 0.7503 | 0.6788 | 0.6815 | 0.6910 | 0.5846 | 0.6427 | 0.6573 |
| | UCCH | 0.7343 | 0.7342 | 0.7410 | 0.6740 | 0.6812 | 0.6945 | 0.6023 | 0.6258 | 0.6371 |
| | MLCAH | 0.7940 | 0.8050 | 0.8050 | 0.6620 | 0.6730 | 0.6870 | 0.5440 | 0.5470 | 0.5940 |
| | DCHMT | 0.8007 | 0.8021 | 0.8065 | 0.6852 | 0.6963 | 0.7009 | 0.6298 | 0.6176 | 0.6616 |
| | **AHLR** | **0.8046** | **0.8052** | **0.8154** | **0.6952** | **0.7040** | **0.7144** | **0.6451** | **0.6557** | **0.6672** |

$\mathbf{S}_{ij}$ represents the cosine similarity between the i-th image and the j-th text embedding. $\mathbf{X}_1^i \cdot \mathbf{X}_2^j$ represents the dot product between vectors, $\left\|\mathbf{X}_1^i\right\|_2$ and $\left\|\mathbf{X}_2^j\right\|_2$ are the L2 norms of $\mathbf{X}_1^i$ and $\mathbf{X}_2^j$, respectively, representing the length of the vector. Since $\mathbf{X}_1$ and $\mathbf{X}_2$ have been normalized, $\mathbf{S}_{ij}$ can be simplified as follows:

$$\mathbf{S}_{ij} = \mathbf{X}_1^i \cdot \mathbf{X}_2^j \tag{26}$$

The labeling matrix $\mathbf{A}$ marks whether each pair of image and text is a positive sample or a negative sample, where a positive sample refers to a matching pair between an image and the corresponding text, and a negative sample refers to a pair between an image and an irrelevant text. The loss between positive samples is calculated as follows:

$$\mathcal{L}_{positive} = \sum_{i=1}^{N}\sum_{j=1}^{N} \mathbf{A}_{ij} \cdot (1 - \mathbf{S}_{ij}) \tag{27}$$

where $\mathbf{A}_{ij}$ is the element in the label matrix, and then the loss of the negative sample pair is calculated. $\xi$ controls the similarity between negative samples and pushes the similarity between negative samples down to the set boundary $\xi$. The specific process is shown as follows:

$$\mathcal{L}_{negative} = \sum_{i=1}^{N}\sum_{j=1}^{N} (1 - \mathbf{A}_{ij}) \cdot \text{ReLU}(\mathbf{S}_{ij} - \xi) \tag{28}$$

## 3 Experiments

In this section, we assess the performance of the proposed AHLR framework across three benchmark datasets using cutting-edge techniques. We subsequently conduct an in-depth analysis through systematic ablation studies to examine the role of each component within our model.

### 3.1 Experimental Settings

In this paper, we use the Vision Transformer as the feature extractor of images and adopt the BPE method to perform word segmentation on text. Subsequently, Transformer is further used to fuse and process the feature representations of images and text. At the same time, the model introduces a dynamic attention mask mechanism. We experimentally analyzed the parameters involved and selected the most appropriate value, which is 1. In the experiment, the batch size is set to 64, the Adam optimizer[15] is used for the main optimization, and the method of dynamically adjusting the learning rate is adopted, where the initial learning rate of 1e-3, a decay schedule of 0.9 times the learning rate every 5 epochs, and a weight decay of 0.2. The AHLR method is mainly implemented

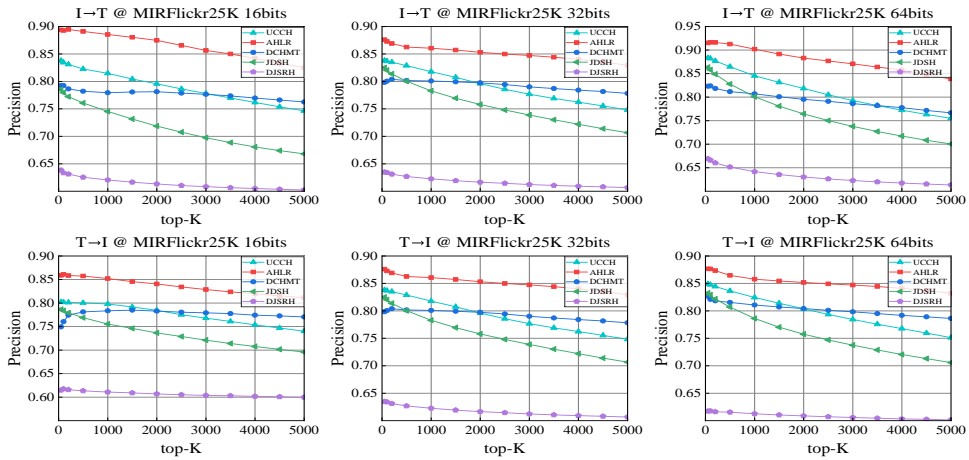

Figure 2: The Top-K curves on the MIRFlickr-25K

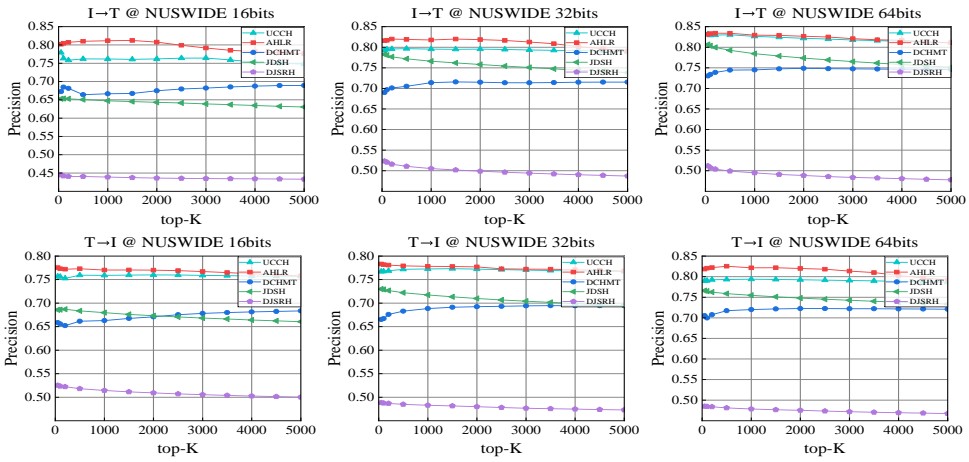

Figure 3: The Top-K curves on the NUS-WIDE

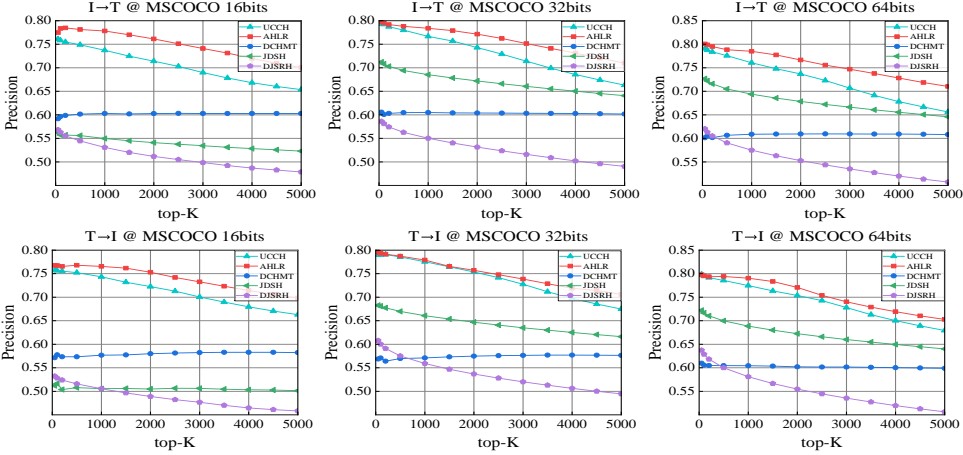

Figure 4: The Top-K curves on the MS-COCO

Table 2: Ablation study on the three datasets

| Task | Method | MIRFlickr-25K | | | NUS-WIDE | | | MS-COCO | | |
|------|--------|--------|--------|--------|--------|--------|--------|--------|--------|--------|
| | | 16bits | 32bits | 64bits | 16bits | 32bits | 64bits | 16bits | 32bits | 64bits |
| I→T | AHLR-M | 0.8086 | 0.8161 | 0.8193 | 0.6753 | 0.6850 | 0.6946 | 0.6396 | 0.6516 | 0.6642 |
| | AHLR-A | 0.8083 | 0.8115 | 0.8179 | 0.6686 | 0.6818 | 0.6894 | 0.6447 | 0.6578 | 0.6688 |
| | **AHLR** | **0.8203** | **0.8233** | **0.8266** | **0.6777** | **0.6884** | **0.6994** | **0.6454** | **0.6582** | **0.6797** |
| T→I | AHLR-M | 0.7992 | 0.8005 | 0.8093 | 0.6929 | 0.7011 | 0.7132 | 0.6319 | 0.6525 | 0.6656 |
| | AHLR-A | 0.7988 | 0.7995 | 0.8066 | 0.6865 | 0.6974 | 0.7031 | 0.6422 | 0.6540 | 0.6669 |
| | **AHLR** | **0.8046** | **0.8052** | **0.8154** | **0.6952** | **0.7040** | **0.7144** | **0.6451** | **0.6557** | **0.6672** |

based on Pytorch[16], and all experiments are run on a server equipped with an NVIDIA GeForce RTX 3080 graphics card with 40GB RAM to ensure the stability of the experiment.

## 3.2 Comparison Methods

There are two cross-modal hashing retrieval tasks (convert image to text and text to image). We compare the performance of the AHLR method with six other cross-modal hashing methods on three datasets. The specific methods are as follows: DCHMT[17], JDSH[18], MLCAH[19], UCCH[20], CDTH[8] and DJSRH[21].

## 3.3 Performance Comparison

To demonstrate the efficacy of the AHRL algorithm, we performed a comparative analysis of its performance across three datasets. Table 1 shows the mAP values of the AHLR method on the MIRFlickr-25K, NUS-WIDE, and MS-COCO datasets. The best results in the table are in bold, where "I → T" denotes image-to-text retrieval and "T → I" indicates text-to-image retrieval.

From the mAP of the three datasets, we can see that the AHLR method we proposed can achieve good performance on both small-scale and large-scale datasets. Compared to the baseline, our method achieves over 10% higher mAP across the three datasets than the lowest experimental method. This is due to the introduction of the auxiliary hash module, which enables the model to generate hash codes containing rich semantic information. When facing larger datasets, the performance between different bits gradually stabilizes. This is because the dynamic mask mechanism proposed by AHLR enables the model to adaptively focus on key parts while ignoring unimportant information, thereby improving retrieval accuracy.

Figures 2, 3 and 4 show the Top-K accuracy curves of the AHLR method on the three datasets. The value of K covers multiple retrieval ranges, mainly including 50, 100, 200, 500, 1000, 1500, 2000, 2500, 3000, 3500, 4000, 4500 and 5000. As can be seen from the figure, as the number of retrieval samples K increases, the retrieval performance of the AHLR method is always significantly higher than that of other comparison methods. As the number of retrieval samples continues to increase, the retrieval performance of the model gradually stabilizes, indicating that our method maintains good stability and reliability while improving retrieval accuracy.

## 3.4 Ablation Study

In order to verify the effectiveness of each module in the model, the following ablation experiments are designed:

- AHLR-M: This experiment removes the dynamic attention mask, and all other modules are the same as in AHLR.
- AHLR-A: This experiment removes the auxiliary hash learning module, and all other modules are the same as in AHLR.

The results of the ablation experiment are shown in Table 2. After AHLR-M removes the dynamic attention mask, the performance of the model in the image-text cross-modal retrieval task is significantly reduced, especially in the accuracy mAP indicator. This shows that the dynamic attention mask

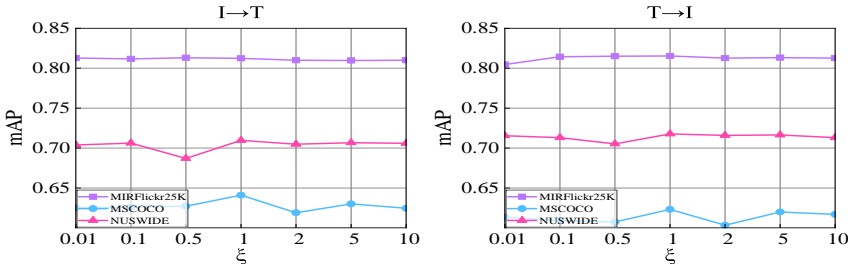

Figure 5: Parameter sensitivity of $\xi$

can effectively capture the key information in the image and text, and flexibly weight it according to the sequence length, thereby improving the representation ability of the feature.

After AHLR-A removes the auxiliary hash learning module, the retrieval performance of the model is reduced, especially under the condition of short hash code length, the performance decline is more obvious. This shows that the auxiliary hash learning module helps to generate more compact and more discriminative hash codes by adaptively weighting the features of each channel, thereby improving the retrieval effect.

The results of the ablation experiment show that the dynamic attention mask and auxiliary hash learning module have an impact on the overall performance of the model and demonstrate their important role in improving cross-modal retrieval performance.

## 3.5 Parameter Sensitivity

We analyze the parameter $\xi$ we designed and select a unified 64-bit hash code to verify it on three data sets. The experimental results show that when the value of the parameter $\xi$ is adjusted, the mAP result will fluctuate accordingly. The specific results are shown in Figure 5. It can be seen from the figure that when the value of $\xi$ is 1, the mAP value of the model is the highest, indicating that the retrieval performance of the model is optimal at this time. As the value of $\xi$ continues to increase, the performance of the model shows a downward trend, indicating that an excessively large $\xi$ may affect the model's expression of feature information, resulting in a decrease in retrieval accuracy. Therefore, based on the comprehensive consideration of the performance of the model, taking $\xi$ as 1 is more in line with the performance of the model, and can ensure good stability and effectiveness on the three data sets.

## 4    Conclusion

We propose an auxiliary hash learning (AHLR) for cross-modal retrieval methods. By introducing a dynamic mask mechanism, the key information between different modalities is automatically selected and weighted to enhance the feature representation and semantic alignment between modalities. In addition, an auxiliary hash layer is constructed to adaptively weight the features of each channel, and combined with the contrast loss function, AHLR can minimize the distance between similar samples and maximize the distance between heterogeneous samples, thereby improving the distinguishing ability of hash codes in cross-modal retrieval tasks and further improving the accuracy of retrieval. Comprehensive experiments have proved the effectiveness of this method. We mainly study the retrieval between images and texts, and cannot effectively process other types of multimodal data (such as video, audio, etc.), which leads to limited applicability of the model and low scalability. Therefore, how to explore semantic alignment methods between multiple different modalities, realizing mutual retrieval, and improving the versatility and applicability of the model is an important direction for future research. Cross-modal hashing retrieval can improve information retrieval efficiency and promote cultural communication and educational innovation. However, it may also leak privacy, threaten security, amplify social bias, and impact employment structure. Therefore, strengthening data privacy security is crucial to ensure its positive social impact.

## Acknowledgments

This work is supported by the Science and Technology Project of Hebei Education Department (No. CXY2024050), the Joint Fund Key Program of the National Natural Science Foundation of China (No. U23B2029), the Fundamental Research Funds for the Central Universities (No. CUC25SG013), the Beijing Municipal Natural Science Foundation (No. L257023), the Agricultural Scientific and Technological Achievements transformation Funds of Hebei Province (No. 2025JNZ-S24), and the North China University of Technology 2025 Youth Research Special Project (No. 2025NCUTYRSP012).

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

# Appendix / supplemental material

## A  Related Work

In this section, we introduce existing work related to our study, focusing on two forms of cross-modal hashing retrieval, namely, unsupervised cross-modal hash retrieval and supervised cross-modal hashing retrieval.

### A.1  Supervised Cross-Modal Hashing

Supervised cross-modal hashing retrieval[22][23][24][25] optimizes the generation of hash codes by leveraging labels or semantic information, thereby improving the accuracy and efficiency of retrieval. The core of supervised cross-modal hashing retrieval lies in mapping data of different modalities (such as images and text) into a shared Hamming space, so that the similarity between different modalities can be measured by the Hamming distance. In the early stage of research, non-deep cross-modal hashing methods were proposed to bridge the semantic gap between different modalities. For example, Semantics-Preserving Hashing (SePH)[26] takes the semantic affinity of given training data as supervised information and converts it into a probability distribution and learns the nonlinear projection of features by minimizing KL divergence to make hash codes for each view.

Deep learning methods have gradually become the mainstream methods for supervised cross-modal hashing retrieval due to their advantages in feature extraction and nonlinear representation. For example, Deep Cross-Modal Hashing (DCMH)[27] integrates feature extraction and hash code learning into a deep convolutional neural network framework to achieve end-to-end learning. Semantic Decomposition and Enhancement Hashing (SDEH)[28] improves the performance of cross-modal retrieval by making full use of multi-label semantic information. This method decomposes the shared semantic information between different modalities and bridges the feature and semantic gap between different modalities, thereby achieving efficient cross-modal retrieval. We adopt this supervised method to fully utilize semantic information to optimize hash code generation and enhance cross-modal retrieval performance.

### A.2  Unsupervised Cross-Modal Hashing

Compared to supervised hashing methods, unsupervised hashing methods[29][30][31][32] do not require labels or semantic information, and map data from different modalities to a shared Hamming space through learning. Unsupervised hashing methods do not require pre-annotated semantic labels but instead discover common representations between different modalities by mining potential intra-modal and inter-modal connections. Early unsupervised hashing methods were mainly based on shallow models, learning hash codes, and hash functions by designing efficient algorithms. These methods usually process manually extracted features. For example, Inter-media hashing (IMH)[33] converts multimedia data from heterogeneous data sources into a common Hamming space and enables a fast search through XOR and bit counting operations.

Deep learning methods have also been widely used in unsupervised hashing. The Unsupervised Contrastive Multi-modal Fusion Hashing Network (UCMFH)[32] uses the pre-trained CLIP model to extract features and enhances the interaction between modalities through the multimodal fusion transformer encoder and contrast loss. Recent unsupervised deep hashing methods pay more attention to the construction of similarity matrices to reduce redundant information and capture the potential associations between modalities. Deep Joint-Semantics Reconstructing Hashing (DJSRH)[21] constructs a joint semantic affinity matrix that integrates the original neighborhood information from different modalities to capture the potential associations of input multimodal instances.

## B  Datasets

To evaluate the proposed method, we choose to conduct experiments on three widely used cross-modal datasets: MIRFlickr-25K[34], NUS-WIDE[35] and MS-COCO[36]. As shown in the table 3, the detailed description of the datasets is as follows.

MIRFlickr-25K contains 25,000 images covering 24 common categories. Each image is accompanied by multiple text labels and is annotated with at least one of the categories. We select image-text pairs

with at least 20 labels as experimental data, randomly extract 2,000 pairs as query sets, the rest as retrieval sets, and randomly select 10,000 pairs from the retrieval sets as training sets.

NUS-WIDE contains 269,648 web images with text labels covering 81 categories. Each image is annotated with at least one of the 81 categories. We select 186,577 image-text pairs, all of which belong to the 10 most common classes, randomly select 2,100 pairs from the dataset as query sets, the rest as retrieval sets, and randomly select 10,500 pairs from the retrieval sets as training sets.

MS-COCO contains 123,289 images covering 80 categories. Each image is labeled as at least one of the 80 categories. In our experiments, we deleted the samples that did not contain valid instances in the text samples to improve the data quality and reliability of the experiment. 5,000 pairs were randomly selected from the dataset as the query set, the rest as the retrieval set, and 10,000 pairs were randomly selected from the retrieval set as training data.

Table 3: Datasets Statistics

| Dataset | Size | Label | Query | raining |
|---|---|---|---|---|
| MIRFlickr-25K | 25,000 | 24 | 2,000 | 10,000 |
| NUS-WIDE | 269,648 | 81 | 2,100 | 10,500 |
| MS-COCO | 123,289 | 80 | 5,000 | 10,000 |

## C  Evaluation Metrics

In the experiment, we use two evaluation indicators to evaluate the performance of the model, namely the mean average precision (mAP) and the Top-K precision curve (top-K curve).

