# OpenReview forum: "Dynamic Masking and Auxiliary Hash Learning for Enhanced Cross-Modal Retrieval"
_NeurIPS.cc/2025/Conference — NeurIPS 2025 poster_

### Official Review · Reviewer_pwot · 2025-06-09

**Clarity:** 3
**Significance:** 4
**Originality:** 4
**Rating:** 6
**Confidence:** 4

**Summary:**

This paper proposed a novel hash learning method for hash learning enhanced cross-modal retrieval. The main challenge in hash learning based cross-model retrieval is the mismatch problem among different modalities, thus leading to wrong retrieval. The proposed method contains two components: Dynamic Masking (DM) and Auxiliary Hash Learning (AHLR). The experimental results show that DM and AHLP could successfully work for hash learning enhanced cross-modal retrieval.

**Questions:**

1. Regarding the DM policy, can the mask cause a negative influence, such as slow convergence speed?

2. Can the author report the computation cost for the proposed method? The motivation for this question is that hash learning suffers from the convergent problem. This paper introduces an additional loss. Thus, I guess the proposed method needs extra training epochs, hindering it from being implemented in real tasks.

**Ethical Concerns:**

["NO or VERY MINOR ethics concerns only"]

**Final Justification:**

This paper proposes a novel hash learning method for enhanced cross-modal retrieval. The initial concerns relate to the potential negative impact of the dynamic mask. After the rebuttal, the author clarifies these concerns. I believe this paper meets the standards of NIPS and has the potential to have a broad impact on hash learning. Therefore, I rate it as strong accept.

**Limitations:**

yes

**Quality:**

4

**Strengths And Weaknesses:**

Strengths:

1. The overall method is sound.

2. The proposed method achieves the SOTA results.

3. The writing is clear and easy to follow.

Weaknesses:

1. Some necessary details about the DM strategy are missing. Based on the ablation study, DM is important for the overall method and even important than AHLR. However, this paper only describes the DM strategy with a few contents in Sec. 2.1. The author should discuss more about the DM such as the rate that should be masked for image features, which may further enhance the contribution of this paper.


To sum up, I think this paper is sound and well-written. The proposed method solves a challenge in hash learning. Meanwhile, i did not find any technical flaws. Thus, i rate it as accept.

---

> ### Author Rebuttal · Authors · 2025-07-29
>
> For Weaknesses:
>
> 1.Weakness:[Some necessary details about the DM strategy are missing. Based on the ablation study, DM is important for the overall method and even important than AHLR. However, this paper only describes the DM strategy with a few contents in Sec. 2.1. The author should discuss more about the DM such as the rate that should be masked for image features, which may further enhance the contribution of this paper.]
>
> Response: Thank you for your comments. We agree that the dynamic masking strategy plays a crucial role in our framework and should be described more clearly and in depth. We will add in the final manuscript how the dynamic masking mechanism determines the selected information as key information, i.e., add the following content. The dynamic masking mechanism guides the model to focus on the key information in the sequence by generating a mask matrix based on the input structure. Specifically, we construct a causal attention mask matrix to limit the attention range of tokens at each position in the Transformer, that is, each token can only focus on the current position and its previous position. Although the mask matrix is determined by the length of the input sequence and does not directly contain semantic information, during the training process, the model can model global and local information through the multi-head attention mechanism and feed forward network, thereby gradually learning key information under the mask restriction.
>
> For Question:
>
> 1.Question: [Regarding the DM policy, can the mask cause a negative influence, such as slow convergence speed?]
>
> Response: Thank you for your comments. Our dynamic masking mechanism is designed to help the model select key information. It selectively "masks" parts of image or text features during training, thereby reducing the information available in each forward propagation, which can lead to slower convergence. This is because the mask restricts the flow of information in the attention mechanism and may initially make optimization more difficult. However, by carefully adjusting the mask rate and training schedule, we found that the benefits of using a dynamic attention mask mechanism outweigh the potential disadvantages. In our experiments, models using DM perform better than variants that do not use masks.
>
> 2.Question: [Can the author report the computation cost for the proposed method? The motivation for this question is that hash learning suffers from the convergent problem. This paper introduces an additional loss. Thus, I guess the proposed method needs extra training epochs, hindering it from being implemented in real tasks.]
>
> Response: Thank you for your comments. The training process is on an NVIDIA RTX 3080 GPU with 64-bit hash codes and batch size 64. Our method takes about 5.75 minutes per epoch for training and 2 minutes 38 seconds for inference on MIRFLICKR, about 10.5 minutes per epoch for training and 5 minutes 45 seconds for inference on MS-COCO, and about 11 minutes per epoch for training and 6 minutes for inference on NUS-WIDE, with peak GPU memory usage around 6.9 GB and GPU utilization between 64% and 77%. Despite introducing an additional loss, our proposed model does not require a significant increase in the number of epochs to converge. We will add the computation cost in the final version.

---

> > ### Comment · Reviewer_pwot · 2025-08-04
> >
> > Thanks to the author`s rebuttal. I have carefully checked all the contents. The author addresses my concerns about the potential negative influence of dynamic masks. I think the overall paper is sound and could further benefit the research of the hash learning, which has the potential to have a broad impact. Thus, I consider raising my score to strong accept.

---

### Official Review · Reviewer_ArmY · 2025-06-29

**Clarity:** 4
**Significance:** 3
**Originality:** 3
**Rating:** 5
**Confidence:** 4

**Summary:**

This article proposes a cross-modal retrieval method called AHLR, which automatically filters and weights key information in images and texts through the dynamic mask mechanism, and combines the auxiliary hash layer to adaptively balance the feature weights of each channel, effectively solving the shortcomings of traditional methods in capturing key information and channel processing, and significantly improving retrieval accuracy. The experimental results show that the AHLR method performs well on multiple benchmark datasets, providing a new solution for cross-modal retrieval.

**Questions:**

1. Has a differentiated masking strategy been used for sequence length differences in different modalities, such as images and text?
2. Is there a detailed design for the network structure of the cross-modal fusion module, such as the number of layers and attention heads?
3.How to ensure semantic alignment of different modal features during the fusion process?

**Ethical Concerns:**

["NO or VERY MINOR ethics concerns only"]

**Final Justification:**

The responses addressed all my previous questions and concerns. I suggest acceptance.

**Limitations:**

yes

**Quality:**

3

**Strengths And Weaknesses:**

Strengths:
1. A novel method of dynamic masking and auxiliary hash learning (AHLR) has been proposed, which effectively solves the problems of channel information imbalance and insufficient capture of key information by jointly utilizing dynamic masking and auxiliary hash learning mechanisms.
2. By constructing an auxiliary hash learning layer, the problem of traditional hash layers ignoring importance differences when processing different channel features has been solved, enhancing the ability to capture key information and effectively suppressing noise and redundant information.
3. The experiment was validated on three benchmark datasets, NUS-WIDE, MIRFlickr25K, and MS-COCO, demonstrating the performance of the method.

Weaknesses:
1. There is a typo in the scale of the NUS-WIDE dataset in Table 3, which needs to be corrected.
2. The depth of performance experiment analysis needs to be strengthened, and the reasons for obtaining relevant experimental results need to be analyzed.
3. In practical applications, if there are problems such as noise, missing values, or inaccurate labeling in the input data, it may affect the performance of the model and require additional data cleaning and preprocessing work to ensure data quality.

---

> ### Author Rebuttal · Authors · 2025-07-29
>
> For Weaknesses:
>
> 1.Weakness: [There is a typo in the scale of the NUS-WIDE dataset in Table 3, which needs to be corrected.]
>
> Response: Thank you for your comments. We sincerely apologize for the typographical error regarding the dataset sizes in Table 3. We have revised the table accordingly in the updated manuscript. This correction does not affect the results or conclusions reported in the paper.
>
> 2.Weakness: [The depth of performance experiment analysis needs to be strengthened, and the reasons for obtaining relevant experimental results need to be analyzed.]
>
> Response: Thank you for your comments. We agree that a more in-depth analysis of the experimental results would help improve the paper. We will add in Section 3.1: Compared with the baseline, our method has a mAP that is more than 10% higher than the lowest experimental method on the three datasets, which is mainly attributed to the auxiliary hash learning module, which helps the model generate hash codes rich in semantic clues, thereby enhancing its generalization ability. On larger datasets, the performance difference between different hash code lengths (16/32/64 bits) gradually decreases. This trend suggests that the dynamic mask mechanism enables the model to better capture key areas and suppress irrelevant or noisy features, resulting in more stable retrieval results. We will add more experiment analysis in the final version.
>
> 3.Weakness: [In practical applications, if there are problems such as noise, missing values, or inaccurate labeling in the input data, it may affect the performance of the model and require additional data cleaning and preprocessing work to ensure data quality.]
>
> Response: Thank you for your comments. In practical applications, there may indeed be noise, missing data, or inaccurate labels in the input data. To address this issue, we perform standard preprocessing steps before feeding the dataset into the model, which alleviates these problems to a certain extent. In addition, the dynamic attention masking mechanism we proposed can adaptively filter out features with less information or more noise, thereby providing a certain degree of robustness to noisy and incomplete data. This design further mitigates the negative impact that may be caused by data quality issues.
>
> For Question:
>
> 1.Question: [Has a differentiated masking strategy been used for sequence length differences in different modalities, such as images and text?]
>
> Response: Thank you for your comments. In our approach, we do consider the sequence length differences of different modalities when applying dynamic attention masks. Our dynamic attention mask sets the corresponding mask matrix M according to the length of the image and text sequences, and the mask matrix is adaptively set according to the sequence length after encoding and position embedding. That is, the mask strategy is dynamically applied according to the input length of each modality, thereby achieving differentiated attention control.
>
> 2.Question: [Is there a detailed design for the network structure of the cross-modal fusion module, such as the number of layers and attention heads? ]
>
> Response: Thank you for your comments. When we further use Transformer to fuse and process image and text features respectively, our network architecture is designed in detail. We use stacked Transformer layers and set the number of layers to 12. We use multi-head self-attention, and the self-attention projection dimension is 64, that is, the query, key, and value vector dimensions of each attention head are all 64.
>
> 3.Question: [How to ensure semantic alignment of different modal features during the fusion process?]
>
> Response: Thank you for your comments. We use Transformer to encode images and texts separately to obtain feature representations, and use the learned linear projection matrix to map the two modal features to a shared embedding space of the same dimension, ensuring that image and text features are compared and fused at the same semantic scale. The Transformer module uses a dynamic attention masking mechanism that can adaptively adjust attention calculations for input lengths of different modalities, improve the ability to focus on key semantic information during feature extraction, and avoid interference from irrelevant information. At the same time, before calculating the similarity, we normalize the feature vectors of both the image and text to reduce the difference in feature scales between different modalities, so that cosine similarity can more accurately reflect the semantic relevance between the two. These designs together ensure the semantic alignment of different modalities during the fusion process.

---

> > ### Comment · Reviewer_ArmY · 2025-08-05
> >
> > The responses addressed all my previous questions and concerns. I suggest acceptance.

---

### Official Review · Reviewer_vdAW · 2025-06-30

**Clarity:** 2
**Significance:** 3
**Originality:** 3
**Rating:** 4
**Confidence:** 3

**Summary:**

The paper introduces dynamic Masking and Auxiliary Hash-Learning (AHLR) for cross-modal (image–text) retrieval. A dynamic masking block first filters out redundant visual patches and text tokens, keeping only the most informative ones. An auxiliary hash-learning layer then re-weights channel features before hashing, addressing the imbalance between modalities. Trained with a contrastive loss, the two components jointly yield markedly better retrieval accuracy on several widely-used benchmarks.

**Questions:**

Q1. What are the actual training and inference runtimes (or FLOPs/parameter counts) of AHLR versus DCHMT and UCCH on MS-COCO?

Q2. What are the advantages of the channel weight allocation strategy of the auxiliary hash layer compared to classical methods such as SENet?

Q3. Equation (5) imposes a causal-style mask. How does this pattern select “key” tokens? Would a learnable top-k pruning score be superior?

Q4. Could the dynamic mask be reused for other multimodal tasks (classification, generation)? What modifications would be required?

I would consider to give a higher score if the authors could address my concerns in Weaknesses and Questions.

**Ethical Concerns:**

["NO or VERY MINOR ethics concerns only"]

**Final Justification:**

Thank you for the detailed responses, which have addressed all my concerns. After reading the comments from other reviewers, I believe this is a good work and hope it will be accepted.

**Limitations:**

Yes

**Quality:**

3

**Strengths And Weaknesses:**

**Strengths**

+ The auxiliary hash-learning (AHL) layer assigns data-dependent weights to each channel, generating more discriminative hash codes and improving cross-modal retrieval precision.

+ Dynamic masking, paired with a contrastive loss, automatically highlights key regions/tokens and enforces tighter semantic alignment between images and text.

+ Experiments on three public datasets show consistent, often sizeable, gains over strong baselines. Furthermore, ablations isolate the contribution of each module.

** Weaknesses**

- Dynamic masks plus AHL add parameters and operations, yet no analysis of training/inference latency or memory use is provided.

- While addressing channel information imbalance and insufficient key information capture is a novel perspective in hash learning, there are some literatures adopting similar thoughts. For example, [A, B] model the relevance between visual and textual modalities, and [C] addresses modality imbalance across modalities. It would strengthen the paper to draw clearer connections with these related works and explicitly clarify the differences. [A] Noise-robust Vision-language Pre-training with Positive-negative Learning. [B] Cross-modal Retrieval with Noisy Correspondence via Consistency Refining and Mining. [C] Test-time Adaption against Multi-modal Reliability Bias.

- The paper states that image and text features are fused via a Transformer, but omits the exact token construction, positional encoding, and cross-attention setup—making reproduction difficult.

---

> ### Author Rebuttal · Authors · 2025-07-29
>
> For Weaknesses:
>
> 1. Weakness: [Dynamic masks plus AHL add parameters and operations, yet no analysis of training/inference latency or memory use is provided.]
>
> Response: Thank you for your comments. The training process is on an NVIDIA RTX 3080 GPU with 64-bit hash codes and batch size 64, our method requires about 5.75 minutes per epoch for training and 2 minutes 38 seconds for inference on MIRFLICKR (5,120 queries over 19,461 gallery items) with peak GPU memory usage of 6.95 GB and 77% GPU utilization; about 10.5 minutes per epoch for training and 5 minutes 45 seconds for inference on MS-COCO (5,120 queries over 117,098 gallery items) with peak GPU memory usage of 6.87 GB and 64% GPU utilization; and about 11 minutes per epoch for training and 6 minutes for inference on NUS-WIDE (5,120 queries over 190,714 gallery items) with peak GPU memory usage of 6.87 GB and 77% GPU utilization.
>
> 2.Weakness: [While addressing channel information imbalance and insufficient key information capture is a novel perspective in hash learning, there are some literatures adopting similar thoughts. For example, [A, B] model the relevance between visual and textual modalities, and [C] addresses modality imbalance across modalities. It would strengthen the paper to draw clearer connections with these related works and explicitly clarify the differences. [A] Noise-robust Vision-language Pre-training with Positive-negative Learning. [B] Cross-modal Retrieval with Noisy Correspondence via Consistency Refining and Mining. [C] Test-time Adaption against Multi-modal Reliability Bias.]
>
> Response: Thank you for your comments. We note that methods such as [A], [B], and [C] use similar ideas in modeling visual-textual correlation or dealing with modality imbalance, but our method is somewhat different from these works in terms of specific mechanisms: [A] Under the synergy of four pre-training objectives and the positive-negative contrast learning mechanism, it dynamically filters out noise correspondences, thereby capturing multi-level visual-textual correlations from token-level, pair-level, intra-modal to cross-modal, and modeling fine-grained semantic correlations between visual and textual modalities, but it is only applicable to large-scale pre-training stages and is not designed specifically for hash code learning; [B] The image-text pairs are divided into three categories and two mechanisms are designed to reconstruct their consistency. By "refining and correcting pseudo-matches in positive samples" and "mining hidden consistency from negative samples", the semantic consistency between images and texts is dynamically reconstructed in the feature spaces of the two modalities, thereby modeling the correlation between visual and textual modalities, but it focuses on robust retrieval scores rather than compact binary hashing; [C] Adaptive attention modules are used to achieve reliable cross-modal fusion, and confidence-aware loss functions are used to achieve robust adaptation, converting modal imbalance into "reliability weights" that can be detected online and corrected in real time, thereby solving cross-modal imbalance without relying on source domain labels, but these mechanisms are not integrated into the training of compact hash representations. Our method focuses on the problem of low matching accuracy and failure to capture key information caused by information redundancy in cross-modal hash learning. We designed a dynamic attention masking module to adaptively filter key cross-modal semantics in the image-text fusion stage, significantly alleviating modal interference and feature redundancy; at the same time, an auxiliary hash layer is introduced and the hash compactness of semantically consistent samples and the separability of heterogeneous samples are enhanced through contrast loss, thereby improving the discriminability and robustness of cross-modal hash retrieval from the source. We will add the above comparison in the related work in the final version.
>
> 3.Weakness: [The paper states that image and text features are fused via a Transformer, but omits the exact token construction, positional encoding, and cross-attention setup—making reproduction difficult.]
>
> Response: Thank you for your comments. In our model, images are divided into fixed-size patches, and positional encoding and learnable class embedding are performed through ViT before being fed into the image transformer; text is encoded through BPE and embedded into a token sequence, and positional encoding is also added before being fed into the text transformer. Image and text tokens are processed by independent transformer encoders, and are not explicitly fused at the token level. The transformer structure consists of multiple layers of multi-head self-attention and feedforward MLP, each layer of which contains residual connections and normalization functions. We will add the explanation in the final version.
>
> For Question:
>
> 1.Question: [What are the actual training and inference runtimes (or FLOPs/parameter counts) of AHLR versus DCHMT and UCCH on MS-COCO?]
>
> Response: Thank you for your comments. We report actual training and inference runtimes on the MS-COCO dataset under the same experimental setup (NVIDIA RTX 3080 GPU, 64-bit hash codes, batch size 64):
>
> UCCH trains in 46-51 seconds per epoch, with per-step forward-backward time of 22-28 ms. Its peak GPU memory usage is 3.6 GB, and inference takes only 2-3 seconds per epoch.
>
> DCHMT requires significantly more time, with training taking 18 minutes per epoch, and each forward-backward batch taking 230-240 ms. Its inference time for full validation (10,240 queries) is approximately 6 minutes, with a peak GPU memory usage of 13.7 GB.
>
> AHLR trains in approximately 10.5 minutes per epoch, with inference time of 5 minutes 45 seconds over 5,120 queries and 117,098 gallery items. Peak GPU memory usage is 6.87 GB, and GPU utilization reaches 64% during training.
>
> 2.Question: [What are the advantages of the channel weight allocation strategy of the auxiliary hash layer compared to classical methods such as SENet?]
>
> Response: Thank you for your comments. SENet uses global average pooling and then two fully connected layers to recalibrate channel importance, but it is mainly used for classification tasks and does not consider the unique semantics of cross-modal retrieval. The auxiliary hashing layer in our model is different from classic methods such as SENet in that it focuses on cross-modal hashing tasks. Instead of using a general channel recalibration, it calculates the channel weights of image and text features separately after dimensionality reduction using the auxiliary hashing layer, and uses a fully connected layer to calculate the importance weight of each channel, allowing the model to capture the nonlinear relationship between channels. Our strategy can emphasize information channels, thereby enhancing discriminative hashing learning.
>
> 3.Question: [ Equation (5) imposes a causal-style mask. How does this pattern select “key” tokens? Would a learnable top-k pruning score be superior?]
>
> Response: Thank you for your comments. We construct a causal attention masking matrix (as shown in Equation (5)) to limit the attention scope of the token at each position in the Transformer, that is, each token can only focus on the current position and its previous position. Although the masking matrix is determined by the length of the input sequence and does not directly contain semantic information, during training, the model can model global and local information through a multi-head attention mechanism and a feedforward network, thereby gradually learning key information under masking restrictions. Regarding alternatives to learnable Top-k pruning scores, we acknowledge that this mechanism can provide a more flexible and explicit way to select semantically important tokens. However, it generally increases computational complexity and may introduce instability during training[1]. Since it is necessary to learn and optimize additional gating variables (Hard-Concrete) and add additional parameters for training, which increases the training time and backpropagation complexity.
>
> REFERENCES
>
> [1]Wang, Tiannan, et al. "Efficientvlm: Fast and accurate vision-language models via knowledge distillation and modal-adaptive pruning." arXiv preprint arXiv:2210.07795 (2022).
>
> 4.Question: [ Could the dynamic mask be reused for other multimodal tasks (classification, generation)? What modifications would be required?]
>
> Response: Thank you for your comments. The dynamic masking mechanism we proposed aims to adaptively focus on key information by generating a mask matrix based on the input sequence structure. This mechanism can indeed be extended to other multimodal tasks, such as classification and generation. For classification tasks, dynamic masking can help the model selectively focus on the most relevant regions or markers in different modalities, and subtle modifications may include adjusting the mask generation criteria to adapt to the input format. For generation tasks, models usually need to follow the sequential generation principle of autoregression, so dynamic masking should continue to maintain or strengthen the restrictions of causal masks to ensure the correct sequential dependencies and information flow during generation. In summary, although dynamic masking can be used for other multimodal tasks, it needs to be adjusted according to the specific task, especially the mask generation logic needs to be modified according to the structure and length of different input modalities.

---

### Official Review · Reviewer_tjNt · 2025-06-30

**Clarity:** 4
**Significance:** 3
**Originality:** 3
**Rating:** 5
**Confidence:** 4

**Summary:**

The paper focuses on research in the field of cross-modal hash retrieval and proposes a novel AHLR method aimed at effectively enhancing feature extraction and alignment capabilities, generating more detailed hash codes, and improving the accuracy of cross-modal hash retrieval. This topic has important research value and practical application significance, and the proposed method has a certain degree of innovation.

**Questions:**

1. Please explain how the image and text features are interactively fused at which step. If not, delete the related description in the article.
2. The dynamic masking mechanism can automatically filter and weight key information in images and text, but how does this mechanism determine which information is key?
3. The paper introduces a contrastive loss function to optimize the generation of hash codes. May I ask, what is the design principle of this loss function? How does it ensure that the generated hash code has stronger discriminative power?

**Ethical Concerns:**

["NO or VERY MINOR ethics concerns only"]

**Final Justification:**

Thank you for the author's response. My problems have been properly resolved. I have decided to keep my score.

**Limitations:**

Yes

**Quality:**

3

**Strengths And Weaknesses:**

S1:A well-written academic paper offers is given. It provides a rigorous and evidence-based approach (AHLR) to solving the semantic gap between heterogeneous modalities such as images and text.
S2: AHLR uses Vision Transformer and Transformer-based text encoders for unified cross-modal feature learning.
S3: The dynamic mask filters irrelevant inputs by sequence length, while the auxiliary module adaptively reweights channels for better hash representations.

W1: Section 2.2 mentions "Transformer is used to further fuse and process the feature representation of images and texts", but Section 2.3 treats image and text learning separately.
W2: The innovation of dynamic masking mechanism lies in automatically filtering key information, but its essential difference from existing attention mechanisms is not clear.
W3: Authors mention using an Adam optimizer with “dynamically adjusting the learning rate,” but do not specify the initial learning rate, decay schedule, or weight decay.

---

> ### Author Rebuttal · Authors · 2025-07-29
>
> For  Weaknesses:
>
> 1.Weakness:[Section 2.2 mentions "Transformer is used to further fuse and process the feature representation of images and texts", but Section 2.3 treats image and text learning separately. ]
>
> Response: Thank you for your comments. In Section 2.2, we mentioned that "Transformer is used to further fuse and process the feature representation of images and texts" means that images use Vision Transformer as feature extractor and then are fused through Transformer to obtain the feature representation of images, at the same time, texts use BPE encoding and are fused through Transformer to obtain the feature representation of texts, as shown in Figure 1. This is why we separate image and text learning in Section 2.3. Since our statement is not clear, it has caused ambiguity that images and texts are fused. We will add more explanations for the utilization of Transformer for images and texts in the final version.
>
> 2.Weakness: [The innovation of dynamic masking mechanism lies in automatically filtering key information, but its essential difference from existing attention mechanisms is not clear.]
>
> Response: Thank you for your comments. The difference between the dynamic masking mechanism and existing attention mechanisms does need to be further clarified. Traditional attention mechanisms (such as self-attention in Transformer) assign continuous soft weights to all tags or image patches, and even if some information contributes little to the task, it will not be truly discarded. The dynamic masking mechanism we proposed can automatically select semantic key information from image and text modalities during training and weight it, thereby suppressing modal redundancy or irrelevant areas, improving semantic consistency between modalities, enhancing feature alignment effects, and improving retrieval efficiency. We will explicitly emphasize the fundamental difference between this mechanism and traditional attention mechanisms in the final version.
>
> 3.Weakness: [Authors mention using an Adam optimizer with "dynamically adjusting the learning rate", but do not specify the initial learning rate, decay schedule, or weight decay.]
>
> Response: Thank you for your comments. We welcome the opportunity to clarify the optimizer settings we used in our experiments. In our experiments, we used the Adam optimizer with an initial learning rate of 1e-3, a decay schedule of 0.9 times the learning rate every 5 epochs, and a weight decay of 0.2. We will adding the initial values in the Experimental Settings in the final version.
>
>
> For  Questions:
>
> 1.Question: [Please explain how the image and text features are interactively fused at which step. If not, delete the related description in the article.]
>
> Response: Thank you for your comments. As we answered in weakness, we mentioned that "Transformer is used to further fuse and process the feature representation of images and texts" means that images use Vision Transformer as feature extractor and then are fused through Transformer to obtain the feature representation of images, while texts use BPE encoding and are fused through Transformer to obtain the feature representation of texts. We will add the above explanations in the final version.
>
> 2.Question: [The dynamic masking mechanism can automatically filter and weight key information in images and text, but how does this mechanism determine which information is key?]
>
> Response: Thank you for your comments. The dynamic masking mechanism guides the model to focus on the key information in the sequence by generating a mask matrix based on the input structure. Specifically, we construct a causal attention mask matrix (as shown in formula (5)) to limit the attention range of tokens at each position in the Transformer, that is, each token can only focus on the current position and its previous position. Although the mask matrix is determined by the length of the input sequence and does not directly contain semantic information, during the training process, the model can model global and local information through the multi-head attention mechanism and feed forward network, thereby gradually learning key information under the mask restriction. We will adding the above explanation in the final version.
>
> 3.Question: [The paper introduces a contrastive loss function to optimize the generation of hash codes. May I ask, what is the design principle of this loss function? How does it ensure that the generated hash code has stronger discriminative power?]
>
> Response: Thank you for your comments. The contrastive loss function we introduced aims to improve the discriminative ability of hash codes by maximizing the similarity between positive sample pairs and minimizing the similarity between negative sample pairs. The design principle of the contrastive loss function is as follows: As shown in formulas (27) and (28), we use the label matrix A to mark whether the sample pairs are positive samples (images and corresponding texts) or negative samples (images and irrelevant texts). For positive sample pairs, the cosine similarity S is made as close to 1 as possible; for negative sample pairs, the model compresses the similarity values between them through a threshold to ensure that they are far away from the positive sample pairs in the embedding space. Therefore, the contrastive loss function can effectively align semantically related image and text modal representations, while increasing the distance between semantically irrelevant samples, thereby ensuring that the generated hash codes are more discriminative.

---

> > ### Comment · Reviewer_tjNt · 2025-08-04
> >
> > Thank you for the author's response. My problems have been properly resolved. I have decided to keep my score.

---

### Decision · Program_Chairs · 2025-09-17

**Decision:**

Accept (poster)

**Comment:**

This paper presents a novel cross-modal retrieval framework combining dynamic masking and auxiliary hash learning to address key challenges in hash code generation. The method effectively tackles channel information imbalance and insufficient key feature extraction through innovative mechanisms that dynamically filter and weight critical information across modalities. All four reviewers recognized the paper's technical foundation and meaningful contribution to cross-modal retrieval. The authors' thorough rebuttal clarified implementation specifics, provided detailed computational analysis, and demonstrated consistent superiority across multiple benchmarks.